

# LIDER: cell embedding based deep neural network classifier for supervised cell type identification

Yachen Tang, Xuefeng Li and Mingguang Shi

Hefei University of Technology, Hefei, China

## ABSTRACT

**Background**. Automatic cell type identification has been an urgent task for the rapid development of single-cell RNA-seq techniques. Generally, the current approach for cell type identification is to generate cell clusters by unsupervised clustering and later assign labels to each cell cluster with manual annotation.

**Methods**. Here, we introduce LIDER (celL embeddIng based Deep nEural netwoRk classifier), a deep supervised learning method that combines cell embedding and deep neural network classifier for automatic cell type identification. Based on a stacked denoising autoencoder with a tailored and reconstructed loss function, LIDER identifies cell embedding and predicts cell types with a deep neural network classifier. LIDER was developed upon a stacked denoising autoencoder to learn encoder-decoder structures for identifying cell embedding.

**Results**. LIDER accurately identifies cell types by using stacked denoising autoencoder. Benchmarking against state-of-the-art methods across eight types of single-cell data, LIDER achieves comparable or even superior enhancement performance. Moreover, LIDER suggests comparable robust to batch effects. Our results show a potential in deep supervised learning for automatic cell type identification of single-cell RNA-seq data. The LIDER codes are available at https://github.com/ShiMGLab/LIDER.

Corresponding author
Mingguang Shi,
mingguang.shi@hfut.edu.cn

## INTRODUCTION

Recent advances in single-cell RNA sequencing (scRNA-seq) techniques provide an avenue for the revolutionized studies of cellular differentiation (*Rizvi et al., 2017*), cellular plasticity (*Rohlenova et al., 2020*) and cellular heterogeneity (*Buettner et al., 2015*). A major drawback in analyzing scRNA-seq data is batch effect, transcriptional noise, variable sensitivity and lacking in biological explanation (*Lopez et al., 2018*). Although scRNA-seq techniques have been made great progress, automated identification of cell types remains a bottleneck in experimental analysis of scRNA-seq data. Currently, methods to identify cell types usually have generated cell clusters by unsupervised clustering and then assigned labels to each cluster based on manual annotation, but these approaches are generally tedious and time-consuming. Moreover, since clustering is typically unsupervised learning, there is no guarantee that the resulting clusters are biologically meaningful, and manual

annotation-based methods are usually difficult if the researcher does not have sufficient background knowledge to find the best match between the marker genes associated with each cluster and a specific cell type.

A number of computational methods have been developed for scRNA-seq data including the identification of cell types (*Shekhar et al., 2016*), the discovery of single-cell regulatory networks (*Aibar et al., 2017*), the construction of cell lineages from single-cell transcriptomes (*Chen, Rénia & Ginhoux, 2018*), the combinatorial prediction of marker panels from single-cell transcriptomic data (*Delaney et al., 2019*), the prediction of single-cell perturbation responses from single-cell gene expression data (*Lotfollahi, Wolf & Theis, 2019*), and the identification of axes of variation among multicellular biospecimens profiled at single-cell resolution (*Chen et al., 2020b*). These findings also face a variety of challenges due to high-dimensional scRNA profiles. Firstly, single-cell transcriptomics is easily influenced by extreme noise and dropout events. Secondly, computational memory may render these proposed methods poorly scalable for massive single-cell RNA-sequencing data. Several supervised learning methods have been proposed to tackle the challenges in scRNA-seq analysis. ACTINN automatically identify cell types with a neural network in single cell RNA sequencing (*Ma & Pellegrini, 2020*). SingleCellNet was proposed to classify single cell RNA-seq data across platforms and across species (*Tan & Cahan, 2019*). A hierarchical machine learning framework Moana was developed to enable the construction of robust cell type classifiers from heterogeneous scRNA-Seq dataset (*Wagner & Yanai, 2018*). Although these developed approaches show great potential in improving the predicted performance, their performance mainly depend on the chosen reference dataset, the prioritized predictive features and the developed predictive model.

Various deep learning methods have been proposed to present a scalable deep-learning-based approach scScope for the analysis of cell-type composition from single-cell transcriptomics (*Deng et al., 2019*), develop a self-configuring method nnU-Net for deep learning-based biomedical image segmentation (*Isensee et al., 2021*), predict drug efficacy from transcriptional profiles by a deep learning–based efficacy prediction system called DLEPS (*Zhu et al., 2021*), construct chromatin interaction neural network ChINN to predict chromatin interactions from DNA sequences (*Cao et al., 2021*), and accurately identify SNPs and indels in difficult-to-map regions from long-read sequencing by haplotype-aware deep neural networks called NanoCaller (*Ahsan et al., 2021*). Deep learning identifies complex structure in large data sets by using the backpropagation algorithm which aims to indicate how a machine should change its internal parameters that are used to compute the representation in each layer from the representation in the previous layer (*LeCun, Bengio & Hinton, 2015*). It also indicates how a machine should change its internal parameters that are used to discover the representation in each layer from the representation in the previous layer. Also, deep learning models based dimension reduction scVI (*Lopez et al., 2018*), DCA (*Eraslan et al., 2019*), and scVAE (*Grønbech et al., 2020*) usually compress high-dimensional single-cell data to a low-dimensional hidden space and then reconstruct it by using artificial neural networks.

Inspired by the above viewpoints, we propose LIDER, a joint cell embedding and deep neural network classifier for accurately identifying cell types of scRNA-seq data. LIDER

is developed upon a stacked denoising autoencoder by leveraging the expressions of scRNA-seq data. With the learned cell embeddings, LIDER predicts cell types by building a multi-class scRNA-seq classifier. We make a comprehensive evaluation by the proposed deep supervised learning approach for cell type identification. Benchmarked against state-of-the-art methods across eight types of single-cell data, LIDER achieves comparable or even superior enhancement performance and thus shows great potential in cell type identification. LIDER fairly removes batch effects from single-cell datasets introduced by different techniques. This deep supervised learning method will help overcome the technical hurdles and better identify the identity of each cell for single-cell transcriptomic data. Specifically, LIDER is developed upon a stacked denoising autoencoder to learn encoder–decoder structures for identifying cell embedding. LIDER is a deep supervised method to accurately predict cell types with a deep neural network classifier.

## MATERIALS & METHODS

### Datasets

Table 1 illustrated the distribution of cell types from single-cell RNA sequencing datasets for model evaluation of LIDER. We downloaded the gene expression matrix and cell type annotation of mouse cortical data from the Heisberg Group scRNA-seq dataset (https://hemberg-lab.github.io/scRNA.seq.datasets/) (*Zeisel et al., 2015*), those of pancreatic islets dataset from ArrayExpress (https://www.ebi.ac.uk/arrayexpress/experiments/E-MTAB-5061/) (*Segerstolpe et al., 2016*), those from GSE71585 (*Tasic et al., 2018*), those of a PBMC dataset (https://support.10xgenomics.com/single-cell-gene-expression/datasets/2.1.0/pbmc4k) (*Zheng et al., 2017*), those of a human liver dataset from GSE115469 (*MacParland et al., 2018*), those from GSE36552 (*Yan et al., 2013*), those of a mouse bladder cell dataset from the Mouse Cell Atlas project (*Han et al., 2018*), those of colorectal cancer (CRC) from GSE81861 (*Li et al., 2017*), those of peripheral bipolar cell from GSE118480 (*Peng et al., 2019*). The mouse cell atlas datasets Tabula Muris were collected from https://tabula-muris.ds.czbiohub.org. We performed the *Z*-score transformation $\bar{f} = \frac{f - E(f)}{\mathrm{Std}(f)}$ to standardize the gene expression values of each cell for each dataset, where f represented the expression value of each gene, $E(f)$ denoted the mean of each $f$, and $\mathrm{Std}(f)$ represented the standard deviation of each f respectively. This process made the expression levels comparable across genes.

### Overview of LIDER development and evaluation workflow

Figure 1 illustrates the overview of the supervised cell type identification for single-cell transcriptomic data. LIDER aims to improve the cell type identification of single cell data by using a combination of stacked denoising autoencoder and deep neural network classifier. The cell embeddings based on stacked denoising autoencoders are used for developing deep neural network classifier (Fig. 1A). The low-dimensional representation by training the first level denoising autoencoders (DAE1) is utilized to train the second level denoising autoencoders (DAE2) and this process is repeated with several times to develop finally stacked denoising autoencoders (SDAE) (Fig. 1B). The process of LIDER for cell type identification could be divided into three main steps: (i) the standardized

**Table 1   A summary of single-cell RNA sequencing datasets used for the development and validation of LIDER.**

| Dataset | Number of cells | Number of genes | Numbei of cell types | Sequencing platform | Reference |
|---|---|---|---|---|---|
| Zeisel | 3005 | 19972 | 9 | Illumina HiSeq | *Zeisel et al. (2015)* |
| Segerstolpe | 3514 | 26271 | 15 | Smart-Seq2 | *Segerstolpe et al. (2016)* |
| Tasic | 1474 | 24057 | 8 | Illumina HiSeq | *Tasic et al. (2018)* |
| PBMC | 4340 | 33694 | 8 | 10X | *Zheng et al. (2017)* |
| Macparland | 8444 | 20007 | 11 | Illumina HiSeq | *MacParland et al. (2018)* |
| Yan | 90 | 20214 | 7 | Drop-seq | *Yan et al. (2013)* |
| Mouse | 2100 | 20670 | 16 | Microwell-seq | *Han et al. (2018)* |
| CRC | 363 | 17267 | 6 | Illumina HiSeq | *Li et al. (2017)* |
| Macaque | 30302 | 36162 | 12 | Drop-seq | *Peng et al. (2019)* |
| Tabula Muris (10X) | 35166 | 19371 | 12 | 10X | / |
| Tabula Muris (SS2) | 20946 | 22995 | 12 | Smart-Seq2 | / |

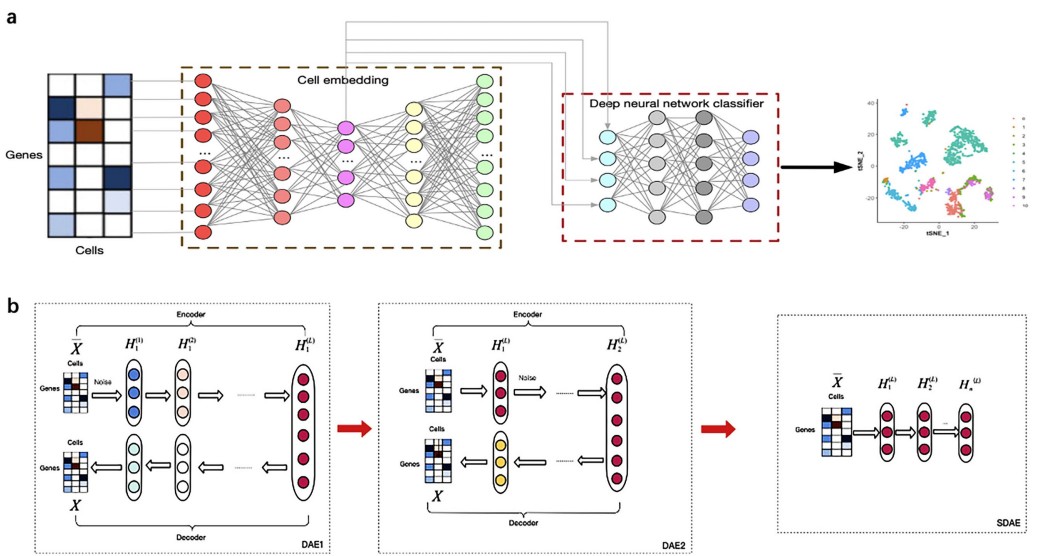

**Figure 1   Building a multi-class scRNA-seq classifier with stacked denoising autoencoder and deep neural network classifier.** (A) ScRNA-seq data are collected and $z$-score transformed. LIDER generates cell embeddings using stacked denoising autoencoder. A deep neural network classifier is then developed by using Adam algorithm for classification tasks. Finally, the cell types are identified by the developed multi-class scRNA-seq classifier. (B) The development of stacked denoising autoencoders. After training the first level denoising autoencoders (DAE1), the obtained representation is used to train the second level denoising autoencoders (DAE2). Stacked denoising autoencoders (SDAE) are usually developed by multiple stacking layers of denoising autoencoders. DAE represents denoising autoencoder and SDAE represents stacked denoising autoencoder respectively.

single-cell transcriptomic data is used as an input of stacked denoising autoencoders; (ii) the stacked denoising autoencoders for identifying cell embedding; (iii) the development of deep neural network classifier based on a parameter optimization algorithm Adam.

## Stacked denoising autoencoder for identifying cell embedding

Denoising autoencoder was developed to learn a robust representation for partial corruption of the input pattern (*Vincent et al., 2008*). In this analysis, we used denoising autoencoder as feature selection method for identifying the low-dimensional representation called cell embedding. For a given single-cell sample, $\overline{X} \epsilon R^{n \times d}$ represents the gene expression matrix where $n$ is the number of single cells and $d$ is the number of the genes. The output $H^{(l)}$ of encoding layer $l$ is calculated as:

$$H^{(l)} = \phi(W_e^{(l)} H^{(l-1)} + b_e^{(l)}) \tag{1}$$

where $\phi$ is an activation function, $W_e^{(l)}$ is the weight matrix, $b_e^{(l)}$ is bias parameters and $H^{(l-1)}$ represents the output of encoding layer $(l-1)$ respectively. Specifically, $H^{(0)} = \overline{X} + \varepsilon$ where $\varepsilon$ is a given type of noise. We randomly mask some observed input features to zeros to develop the corrupted ones, and thus denoising autoencoders are supposed to learn more robust representations.

The decoded layers could be defined as:

$$H^{(l)} = \phi(W_d^{(l)} H^{(l-1)} + b_d^{(l)}) \tag{2}$$

where $\phi$ is an activation function, $W_d^{(l)}$ is the weight matrix, $b_d^{(l)}$ is bias parameters and $H^{(l-1)}$ represents the output of decoding layer $(l-1)$ respectively. The weight matrix and bias parameters are optimized according to the following loss function

$$L = \frac{1}{n} \sum_{i=1}^{n} \sum_{j=1}^{d} (\overline{X}_{ij} - X_{ij})^2 \tag{3}$$

where $X$ is the reconstructed data as the output of the last layer. As a loss function, we use the mean squared error to implement the denoising autoencoder and optimize the parameters.

Stacked denoising autoencoders are usually developed by multiple stacking layers of denoising autoencoders. After training the first layer of denoising autoencoders, its resulting representation is usually used to train denoising autoencoders of the second level. This multi-layer cycle process develops stack multiple denoising autoencoders for the task of layer-by-layer feature extraction, and eventually makes the features more representative (*Vincent et al., 2010*). A stacked denoising autoencoder is used to learn encoder–decoder structures for identifying cell embedding. The activation function $\text{ReLU} = \max(0, x)$ is used for the encoder and decoder networks except for the bottleneck layer and last decoder layer, in which we use $\tanh(x) = \frac{1-e^{-2x}}{1+e^{-2x}}$ as the activation function. The loss function of stacked denoising autoencoders is defined as the reconstruction loss function Eq. (3) between the input data $\overline{X}$ and the output data $X$. After training each layer by minimizing the reconstruction loss of each layer, all encoder layers are connected with all decoder layers in reverse layer-by-layer training order, in which all parameters are fine-tuned by minimizing the loss. After training the first level denoising autoencoders, the obtained representation is used to train the second level denoising autoencoders. This procedure could be repeated to develop stacked denoising autoencoders for learning and stacking several layers. The embedded features were obtained from the representation $H^L$ of stacked denoising autoencoders (Fig. 1B).
### Deep neural network classifier

Given $\{(x_i, y_i)\}_{i=1}^{n}$, $\{x_i\}_{i=1}^{n}$ represents the embedded features for cell $i$ and $\{y_i\}_{i=1}^{n}$ indicates a cell type label of each cell, where $n$ is the number of single cells. A neural network contains an input layer, two hidden layers and one output layer (Fig. 1A). The number of nodes in the input layer is equal to the number of derived features from stacked denoising autoencoder in a training set. The number of nodes in the output layer is equal to the number of cell types of single-cell transcriptomic data. The neural network could be described as follows (*Chen et al., 2020a*):

$$O_1 = \text{sigmoid}(W_1 X + b_1) \tag{4}$$

$$O_m = \text{sigmoid}(W_m O_{m-1} + b_m), \quad 2 \leq m \leq M \tag{5}$$

$$\bar{y} = \text{softmax}(W_{m+1} O_M + b_{m+1}) \tag{6}$$

where $X$ represents the output from stacked denoising autoencoders, $W_m$ is the weight matrix of the *mth* layer, $b_m$ is the bias vector of the *mth* layer, $O_{m-1}$ is the output of the $(m-1)$th layer, $m$ represents the layer of a neural network and $\bar{y}$ is the output of the classification layer. For the input and hidden layers, we use the sigmoid function as the activation function, which is defined as follows:

$$\text{sigmoid}(x) = \frac{1}{1 + e^{-x}} \tag{7}$$

For the output layer we use the softmax activation function, which is defined as follows:

$$\text{softmax}(x_j) = \frac{\exp(x_j)}{\sum_{j=1}^{J} \exp(x_j)} \tag{8}$$

where $J$ element indicates a total of $J$ cell types in the training set. We define the cross-entropy function as classification loss function measuring the discrepancy between the predicted and true class label:

$$L = -\sum_{i=1}^{n} \sum_{j=1}^{J} y_{ji} \log \bar{y}_{ji} \tag{9}$$

where $y_{ji}$ is the true label of cell type for $x_i$ and $\bar{y}_{ji}$ represents the probability of $x_i$ belongs to the $j$ th cell type. We utilize a multilayer neural network to identify cell types as a supervised-learning problem, that is, to accurately predict the class labels of cell types.

Adam is a promising stochastic optimization algorithm for first-order gradient-based optimization with stochastic objective functions. This approach is invariant to diagonal rescaling of the gradients and computationally efficient with little memory requirements (*Kingma & Ba, 2014*). In this analysis, we use the Adam algorithm to develop deep neural network classifier. $f(\theta)$ is a stochastic objective function with respect to parameters $\theta = (W, b)$ and our aim is to minimize the expected value of the objective function according to the cross-entropy function Eq. (9).

## Experimental setting and LIDER implementation

We implemented the stacked denoising autoencoders using Keras (https://github.com/fchollet/keras) and the code was written in Python. For all single-cell datasets, the encoder network of the stacked denoising autoencoders was set to size d-500-500-2000-1000 for the fully connected multilayer perceptron (MLP) where d represents a number of genes from single-cell transcriptomic data, while the decoder network was a MLP of size 1,000-2,000-500-500-d. In this analysis, the neurons hold 500, 500, 2,000 and 1,000 for each layer in the encoder network and 1,000, 2,000, 500 and 500 for each layer in the decoder network respectively. In the process of pretraining stacked denoising autoencoders, the drop ratio for constructing $i$ th stacked denoising autoencoder was set to 0.2 where drop ratio represented the proportion of random disconnection for dropout, and each denoising autoencoder was trained for 200 epochs during the layer-by-layer pre-training period using the stochastic gradient descent (SGD) algorithm. After pre-training stacked denoising autoencoders, we then trained 400 epochs using the SGD algorithm for overall fine-tuning, and the learning rate was set to 0.1 with decayed 10 times every 80 epochs. In this analysis, the selected features of 1,000 dimension from single-cell data as low-dimensional representations were derived to develop a classifier for identifying cell types.

The deep neural network classifier was implemented by Python using TensorFlow in the process of model development (https://www.tensorflow.org). We designed a four-layer neural network with the number of nodes in the input layer (1000), two hidden layers (528 and 256 respectively), and the number of nodes in the output layer according to the number of categories of the single-dell data. We then set the learning rate to 1e−3, the training period for supervised model initialization to 1200, and the batch size to 256 respectively. Good default settings for the implementation of Adam algorithm are $\alpha = 0.001$, $\beta_1 = 0.9$, $\beta_2 = 0.999$ and $\varepsilon = 10^{-8}$ respectively. In our analysis, we randomly divided the single-cell transcriptomic data into training and test datasets, containing 80% and 20% of the samples respectively. The deep neural network classifier was trained on the training set using the optimal parameters, validated on the test dataset, and evaluated based on accuracy.

## Baseline methods

We compared the performance of LIDER with five baseline methods, such as logistic regression multiclassification algorithm (LR), Moana (*Wagner & Yanai, 2018*), Single-CellNet (*Tan & Cahan, 2019*) and ACTINN (*Ma & Pellegrini, 2020*). Logistic regression multiclassification algorithm (LR) is developed by Python for further comparison. L2 regularization is used in logistic regression modeling. An implementation of Moana is made by Python and found at https://github.com/yanailab/moana. All codes about ACTINN are implemented in Python and available at https://github.com/mafeiyang/ACTINN. The SingleCellNet code by R is available from https://github.com/pcahan1/singleCellNet.

# RESULTS

## LIDER improves prediction performance for cell type identification

To test whether autoencoder based deep learning model could improve prediction performance, we developed LIDER by training classifiers and assessing their performance

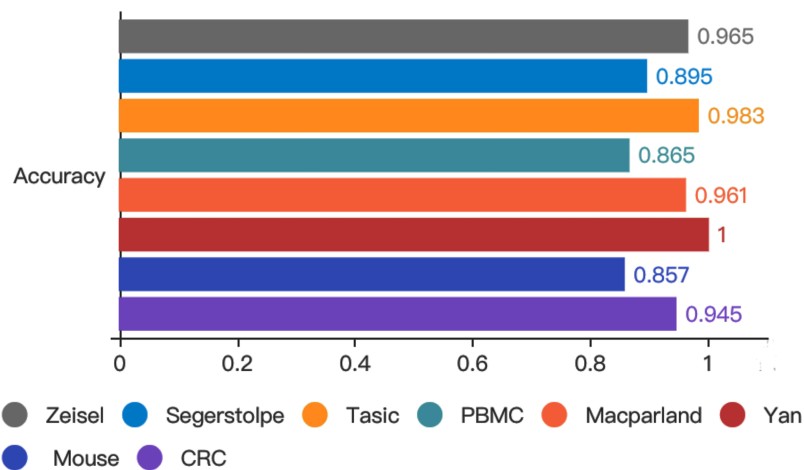

**Figure 2** **Accuracy of validated test dataset from eight real single-cell transcriptomic datasets by us-ing LIDER.** We randomly divide the single-cell transcriptomic data into 80% training dataset and 20% test dataset respectively. The deep neural network classifier is trained on the training set, validated on the test dataset, and evaluated based on accuracy. Zeisel (0.965), Segerstolpe (0.895), Tasic (0.983), PBMC (0.865), Macparland (0.961), Yan (1), Mouse (0.857), CRC (0.945).

when applied to test cohorts. LIDER was used to classify the real single-cell transcriptomic data including Zeisel, Segerstolpe, Tasic, PBMC, MacParland, Yan, Mouse and CRC respectively. The prioritized features based on stacked denoising autoencoders was utilized for the training set and deep neural network classifier was developed to identify cell types of single-cell transcriptomic data. Based on the optimized parameters from training data set, LIDER was developed to classify the test data set of single-cell transcriptomic data. As shown in Fig. 2, the proposed LIDER model achieved an accuracy between 0.857 and 1.0 and an average value of accuracy 0.934 for eight types of single-cell datasets.

To further demonstrate the effectiveness of the method, we compared the proposed LIDER with logistic regression multiclassification algorithm (LR), Moana, SingleCellNet and ACTINN respectively. These four baseline models were trained on the training data set and validated on the test data set. LIDER achieved an accuracy in four test datasets (Figs. 3A–3D), which yielded 1.6%, 4.6%, 10% and 5.8% (Fig. 3A), 2.6%, 5.2%, 7.6% and 6.0% (Fig. 3B), 1.1%, 3.4%,11.7% and 5.2% (Fig. 3C), 1.5%, 2.9%, 5.1% and 4.8% (Fig. 3D) increase in accuracy as compared to LR, Moana, SingleCellNet and ACTINN respectively. An accuracy in four test datasets was yielded by LIDER (Figs. 3E–3H), which was 1.6%, 5.4%, 11.3% and 7.8% (Fig. 3E), 2.9%, 7.5%, 9.8% and 8.9% (Fig. 3F), 1.3%, 4.1%, 7.1% and 5.7% (Fig. 3G), 1.6%, 3.2%, 8.4% and 5.7% (Fig. 3H) higher than that obtained by LR, Moana, SingleCellNet and ACTINN respectively. Overall, independent test results clearly indicated that LIDER could perform better than a diverse panel of cell type identification methods. The performance demonstrated great potential in the deep supervised learning. LIDER converts gene-level expressions into an encoded lower-dimensional representation of single-cell RNA-seq data. The autoencoder removes noise and leaves a high-value

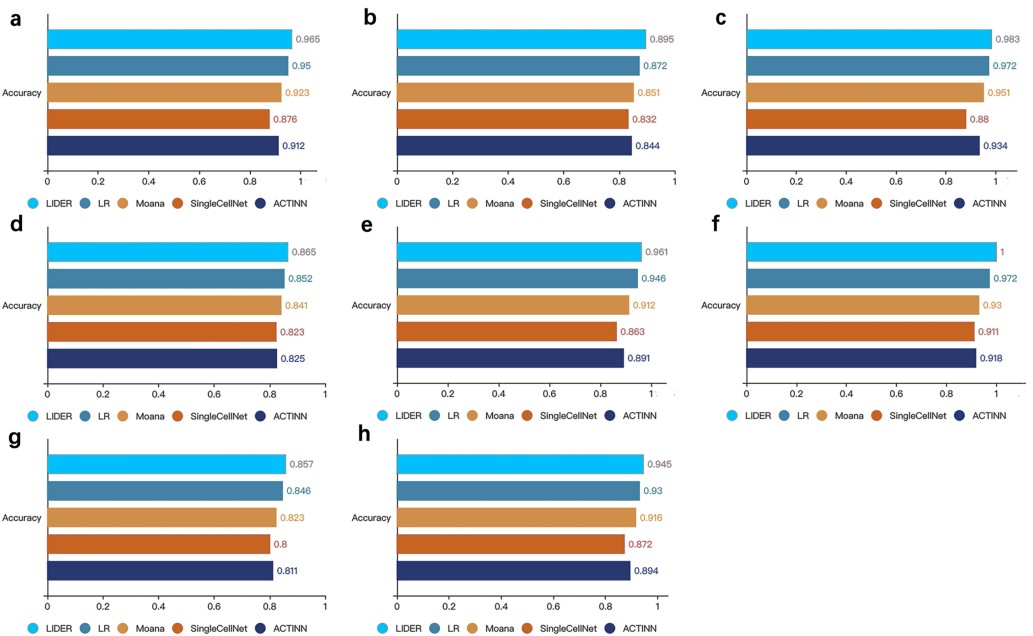

**Figure 3** **LIDER improves prediction performance for cell type identification.** Accuracy of validated test dataset by using LIDER, logistic regression multiclassification algorithm (LR), Moana, SingleCellNet and ACTINN for eight single-cell transcriptomic datasets. In this analysis, 80% training dataset and 20% test dataset are divided from the whole single-cell transcriptomic data. Each subplot represents the accuracy from LIDER and four baseline methods for each single-cell transcriptomic dataset. (A) Zeisel dataset. LIDER (0.965), LR (0.95), Moana (0.923), SingleCellNet (0.876), ACTINN (0.912). (B) Segerstolpe dataset. LIDER (0.895), LR (0.872), Moana (0.851), SingleCellNet (0.832), ACTINN (0.844). (C) Tasic dataset. LIDER (0.983), LR (0.972), Moana (0.951), SingleCellNet (0.88), ACTINN (0.934). (D) PBMC dataset. LIDER (0.865), LR (0.852), Moana (0.841), SingleCellNet (0.823), ACTINN (0.825). (E) MacParland dataset. LIDER (0.961), LR (0.946), Moana (0.912), SingleCellNet (0.863), ACTINN (0.891). (F) Yan dataset. LIDER (1), LR (0.972), Moana (0.93), SingleCellNet (0.911), ACTINN (0.918). (G) Mouse dataset. LIDER (0.857), LR (0.846), Moana (0.823), SingleCellNet (0.8), ACTINN (0.811). (H) CRC dataset. LIDER (0.945), LR (0.93), Moana (0.916), SingleCellNet (0.872), ACTINN (0.894).

representation from the input data. The classifier can perform better because the algorithm is able to learn the patterns in the data from a smaller set of a high-value input.

## LIDER achieves similar cell type identification with the true identity

To investigate the validity of the proposed method LIDER, we compared it with true cell types for the Zeise, Segerstolpe, Tasic, PBMC, MacParland, Yan, and Mouse datasets respectively. We applied principal component analysis (PCA) and t-distributed stochastic neighbor embedding (tSNE) to reduce the dimension and plotted the data points on a two-dimensional plane. A graph-based clustering approach was used to cluster the cells for cell type identification. Figure 4 shows the 2D visualizations of the true cell types and the predicted labels from LIDER. In the identified cell embedding of the proposed method LIDER, cells with the same type are correctly separated with only few outliers (Fig. 4). The vast majority of the predicted cell type from LIDER is similar with the true identity of each

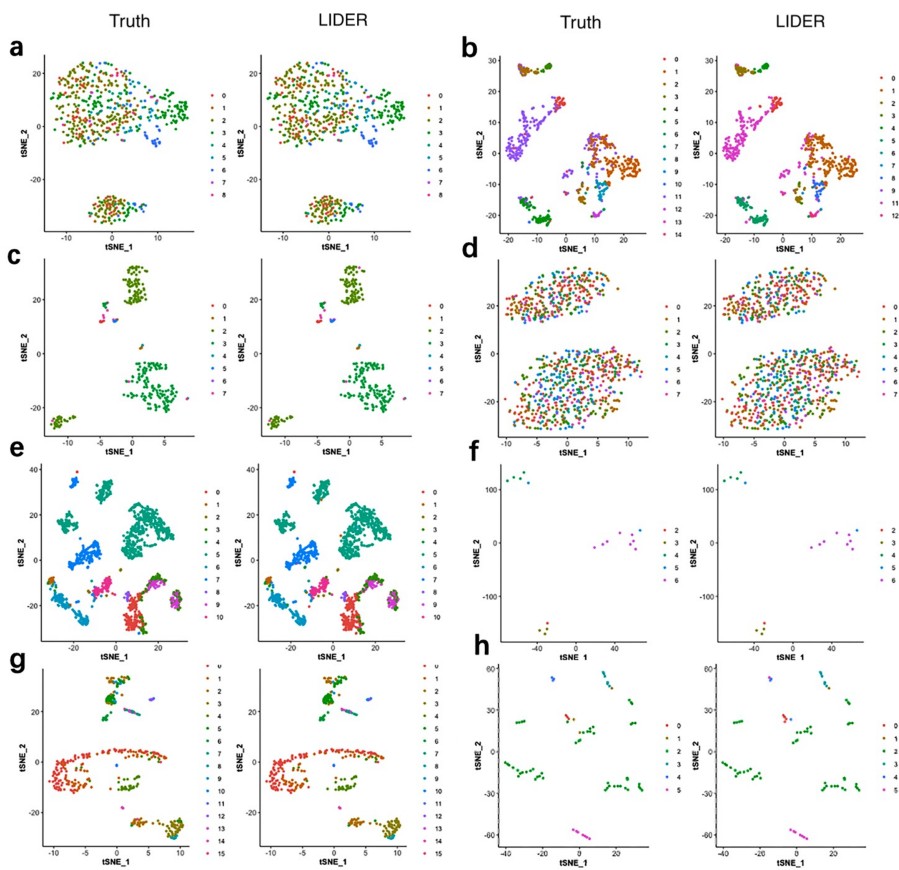

**Figure 4 LIDER achieves similar performance with the true identity of cell type.** Each sub-figure represents the 2D visualizations of the true cell types and the predicted labels from LIDER for eight single-cell transcriptomic datasets respectively. In each sub-figure, the left subplot represents the 2D visualization of the true identity of each cell, and the right subplot represents the 2D visualization of the predicted cell type from LIDER respectively. Each point represents a cell in each sub-figure. (A) Zeisel dataset. (B) Segerstolpe dataset. (C) Tasic dataset. (D) PBMC dataset. (E) MacParland dataset. (F) Yan dataset. (G) Mouse dataset. (H) CRC dataset.

cell from single-cell datasets. These results suggested that deep supervised learning could help improve prediction performance.

## LIDER accurately identifies cell types by using stacked denoising autoencoder

In supervised learning methods, feature selection usually plays an important role in improving classification performance. Therefore, we explore whether stacked denoising autoencoders could improve the prediction performance for cell types identification. As a reference, we used principal component analysis (PCA) based neural network classifier where the architecture of the neural network was similar with the classifier development in LIDER. Specifically, PCA is a popular method for feature selection and shows great potential in dimension reduction of high dimensional data. The PCA algorithm was implemented by Python from PCA package in Sklearn. The performance accuracy is

defined by accuracy $= \frac{1}{n}\sum_{i=1}^{n}\delta(y_i^p, y_i)$, where $\delta = 1$ for $y_i^p = y_i$ and $\delta = 0$ for $y_i^p \neq y_i$ respectively. $y_i^p$ represents the predicted cell type of each single cell and $y_i$ represents the true cell type of each single cell.

Figure 5 illustrated the accuracy between the proposed method LIDER and the PCA-based neural network classifier from the test dataset of eight types of single-cell transcriptomic data. LIDER achieved an accuracy from seven datasets, which was 7.6% (Fig. 5A), 6.3% (Fig. 5B), 1.6% (Fig. 5D), 1.2%(Fig. 5E), 3.8%(Fig. 5F), 5.7% (Fig. 5G), and1.2% (Fig. 5H) higher than that from the PCA-based neural network classifier respectively. Moreover, LIDER achieved an accuracy in the Tasic dataset, which was comparable with that obtained by the PCA-based neural network classifier (Fig. 5C). The results suggested that LIDER was able to identify cell embeddings from the high-dimensional single-cell transcriptomic data by using stacked denoising autoencoder and achieved significantly better performance. Compared with PCA, LIDER developed stack multiple denoising autoencoders which made the selected features more representative of single-cell RNA-seq data.

## LIDER suggests comparable robust to batch effects

Although scRNA-seq studies shows great potential with the characteristic of unprecedented high resolution, a challenging task is to analyze scRNA-seq data with different batches from the generation at different times. Failure to remove batch effects may lead to a false explanation of true biological variations in downstream analysis. To evaluate the performance when reducing the impact of batch effects, we used the Macaque dataset containing animal-level batch effects for further analysis. We still chose 80% as the training set and 20% as the test set. LIDER is effective in removing batch effects and achieves a high accuracy 0.972 when validated on the test data set. To test performance of LIDER accounting for removing batch effects which are due to different techniques, we trained it on single-cell data developed by one platform and tested it on single-cell data developed by another platform. LIDER was trained on the 10X cells and then tested on the SS2 cells. It achieved a testing accuracy of 1.0. Among the seven incorrectly predicted cells, three B cells were predicted as hepatocytes, two monocytes were predicted as stromal cells, and there were remaining two mispredictions. We then trained the proposed method on the SS2 dataset and tested it on the 10X dataset. LIDER obtained a testing accuracy of 0.999. Among the thirty-three incorrectly predicted cells, four B cells were predicted as cardiac muscles, four B cells were predicted as epithelial cells, three epidermises were predicted as natural killer (NK) cells, three epidermises were predicted as hepatocytes, two cardiac muscles were predicted as epithelial cells, two cardiac muscles were predicted as T cells, two epidermises were predicted as epithelial cells, two epidermises were predicted as endothelial cells, two T cells were predicted as B cells, and there were also several mispredictions. The results showed that LIDER fairly removed batch effects from single-cell datasets introduced by different techniques.

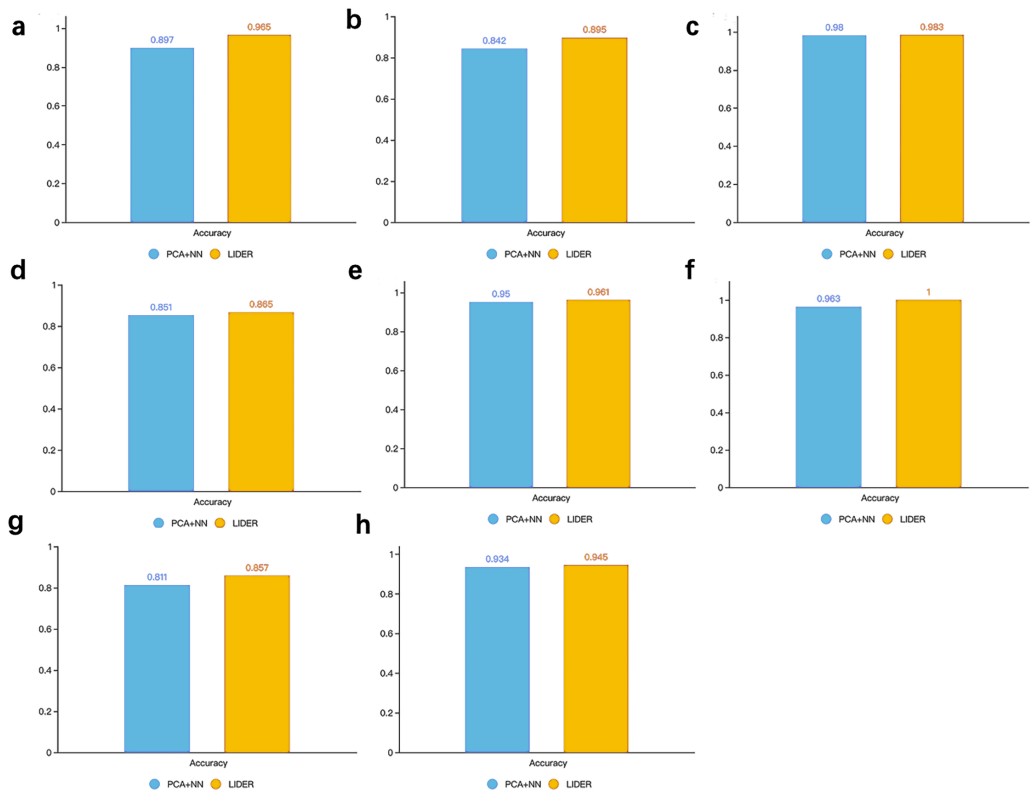

**Figure 5** **Accuracy of validated test dataset by using LIDER and PCA based neural network classifier for eight single-cell transcriptomic datasets.** Each subplot represents the accuracy from LIDER and PCA based neural network classifier for each single-cell transcriptomic dataset. (A) Zeisel dataset. PCA + NN (0.897), LIDER (0.965). (B) Segerstolpe dataset. PCA + NN (0.842), LIDER (0.895). (C) Tasic dataset. PCA + NN (0.98), LIDER (0.983). (D) PBMC dataset. PCA + NN (0.851), LIDER (0.865). (E) MacParland dataset. PCA + NN (0.95), LIDER (0.961). (F) Yan dataset. PCA + NN (0.963), LIDER(1). (G) Mouse dataset. PCA + NN (0.811), LIDER (0.857). (H) CRC dataset. PCA + NN (0.934), LIDER (0.945).

## DISCUSSION

In this study, we present LIDER, a deep computational model for jointly identifying cell embedding and developing deep neural network classifier through scRNA-seq data. Our approach relies on stacked denoising autoencoder to derive cell embeddings which are robust and stable features for matching expression patterns. The deep neural network classifier in LIDER has better predictive performance when compared to baseline methods for accurately identifying the annotation of each cell.

Denoising, referring to cleaning partially corrupted input, suggests that a good representation is able to be robustly developed from a corrupted input and be useful for recovering the corresponding clean input. A denoising criterion is critical for an unsupervised learning to guide the learning of low-dimensional representations for single-cell transcriptomic data (*Vincent et al., 2010*). It is expected that these representations are able to capture useful structure in the input distribution and they are robust and

stable under the corruptions of the input scRNA-seq data. Besides, the pre-training phase of the denoising autoencoder is stacked for each layer, which could improve the training efficiency of a deep leaning model for selecting the relevant features. Specifically, deep learning has attracted much attention because of its ability to extract complex features automatically. Deep neural network has stronger fitting capacity for nonlinear object, and the fitting capacity for nonlinear boundary is raising with the increase of the number of layers. In this article, Adam is a first-order gradient-based optimization method and has been successfully used to train deep neural networks. This algorithm is invariant to diagonal rescaling of the gradients which improves the predictive performance for training a four-layer full-connected neural network. We used a stacked denoising autoencoder to learn encoder–decoder structures for identifying cell embedding. This multi-layer cycle process develops stack multiple denoising autoencoders for the task of feature extraction, which makes the selected features more representative and is critical for batch effects removal of single-cell RNA-seq data. Adam is a promising method with first-order gradient-based optimization of stochastic objective functions for adaptive estimates of lower-order moments (*Kingma & Ba, 2014*). This method is invariant to diagonal rescaling of the gradients and is computationally efficient with little memory requirements. To compare with the proposed method LIDER, we used NN classifier with the raw gene expression data and this classifier achieved an accuracy of 0.902 in Zeisel, 0.84 in Segerstolpe, 0.978 in Tasic, 0.84 in PBMC, 0.951 in MacParland, 0.972 in Yan, 0.801 in Mouse and 0.927 in CRC respectively. LIDER achieved 6.98%, 6.54%, 0.51%, 2.97%, 1.05%, 2.88%, 6.99% and 1.94% increase in accuracy in comparison with NN classifier. The performance demonstrated that the auto-encoder learned features from LIDER could improve the cell type identification.

Single-cell RNA sequencing measures gene expression at the single-cell level and provides a high resolution of cellular differences, which greatly promotes the understanding of cell functions, disease progression, and treatment response (*Gao, 2018*). We applied the scRNA-seq datasets across different platforms to identify cell types. The performance suggest that LIDER is potential for automatic cell type identification of single-cell RNA-seq data. To estimate the performance of independent test dataset, we used scRNA-seq dataset with 22,010 genes × 10,784 cells representing the expression levels of nuclear genes (*Ji et al., 2022*) to validate the method LIDER. We stratified the single-cell dataset into 70% training samples and 30% test samples according to the original sample distribution of each cell type. The gene expression values for each cell were Z-score transformed to make expression level comparable across genes. We processed training and test data sets independently to guarantee the independence of a test dataset. LIDER achieved an accuracy of 0.95 in the training samples and 0.83 in the test samples respectively. Interestingly, transformer-based model has been developed for cell type identification of single-cell RNA-Seq data (*Song et al., 2022*). Recently, a comprehensive and high-performance framework named CIForm is proposed based on the transformer for cell type annotation (*Xu et al., 2023*). CIForm as a deep learning model is structured by four modules including gene embedding, positional encoding layer, transformer encoder and classification layer. An implementation of CIForm is made by Python and found at https://github.com/zhanglab-wbgcas/CIForm. We

normalized and log-transformed gene expression data and selected the top 2000 highly variable genes (HVGs) for single-cell transcriptomic data. CIForm achieved an accuracy of 0.96 in Zeisel, 0.92 in Segerstolpe, 0.95 in Tasic, 0.89 in PBMC, 0.98 in MacParland, 0.87 in Yan, 0.85 in Mouse and 0.92 in CRC respectively. Predicted performance clearly indicated that LIDER could perform better than CIForm for a diverse panel of single-cell data including Tasic, Yan, Mouse and CRC. Moreover, these results suggested that CIForm based on the transformer showed great potential for cell-type annotation of large-scale scRNA-seq data.

Despite the fact that LIDER is potential for accurate cell type identification, the development of deep supervised learning model always requires parameter tuning and model training. For all deep neural network classifiers, the final developed model mainly depends on the hyperparameters used to train the classifier. Moreover, LIDER is a typical supervised learning method and develops the classifier dependent on the quality of the cell type annotations. In addition, advances in stacked denoising autoencoders may facilitate to perform feature selection tasks. Therefore, it would be interesting to investigate deep denoising autoencoders for their ability to develop valuable representations.

## CONCLUSION

In conclusion, a deep supervised learning method LIDER accurately identifies cell types from scRNA-seq data. The stacked denoising autoencoder provided insights with great potential for identifying cell embeddings by matching expression patterns of single-cell transcriptomic data. The deep guided neural network classifier represents a novel approach to leverage cell types with machine learning by building a multi-class scRNA-seq classifier, providing a platform for identifying the type of each cell that may be broadly applicable for single-cell transcriptomic data across different species, conditions, and technologies.

### Funding

This work was supported by grants from the National Natural Science Foundation of China (61572166). The funders had no role in study design, data collection and analysis, decision to publish, or preparation of the manuscript.

### Grant Disclosures

The following grant information was disclosed by the authors:
National Natural Science Foundation of China: 61572166.

### Competing Interests

The authors declare there are no competing interests.

### Author Contributions

- Yachen Tang performed the experiments, analyzed the data, prepared figures and/or tables, and approved the final draft.

- Xuefeng Li analyzed the data, prepared figures and/or tables, and approved the final draft.
- Mingguang Shi conceived and designed the experiments, performed the experiments, analyzed the data, authored or reviewed drafts of the article, and approved the final draft.

## Data Availability

Data and code are available at GitHub: https://github.com/ShiMGLab/LIDER.

Data are originally taken from:

- Heisberg Group scRNA-seq dataset (gene expression matrix and cell type annotation of mouse cortical data): https://hemberg-lab.github.io/scRNA.seq.datasets/.

- ArrayExpress (pancreatic islets dataset): https://www.ebi.ac.uk/arrayexpress/experiments/E-MTAB-5061/.

- GenBank: GSE71585, GSE115469, GSE36552, GSE81861, GSE118480.

- PBMC dataset: https://support.10xgenomics.com/single-cell-gene-expression/datasets/2.1.0/pbmc4k.

- Tabula Muris (mouse cell atlas datasets): https://tabula-muris.ds.czbiohub.org.

## Supplemental Information

Supplemental information for this article can be found online at http://dx.doi.org/10.7717/peerj.15862#supplemental-information.

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
