# Peer review of "LIDER: cell embedding based deep neural network classifier for supervised cell type identification"

_PeerJ, doi:10.7717/peerj.15862_

## Round 0.1 · original submission · Major Revisions

Dear Dr. Tang and colleagues:

Thanks for submitting your manuscript to PeerJ. I have now received two independent reviews of your work, and as you will see, the reviewers raised some concerns about the research and manuscript. Despite this, these reviewers are generally optimistic about your work and the potential impact it will have on research studying automatic cell type identification. Thus, I encourage you to revise your manuscript, accordingly, taking into account all of the concerns raised by both reviewers.

The datasets used for your comparisons seem to not be ideal for these analyses. I strongly suggest using the recommended comparative approaches to robustly evaluate LIDER. Please provide more details on extent of your analyses and ensure that all datasets and results are presented such that your work can be fully evaluated and repeated.

Thus, I encourage you to revise your manuscript, accordingly, taking into account all of the concerns raised by the two reviewers.

Good luck with your revision,

-joe

·

Basic reporting

In the study presented described in the manuscript, these authors introduced a deep supervised
learning method that combines cell embedding and deep neural network classifier for automatic
cell type identification. Although these authors aimed to solve a very important topic in bioinformatics
research filed, they did not conduct very professional work in the present study.

1. I don't think they can use datasets using different technologies in one task. the authors
need read some papers to understand each library construction technologies, particularly 10X
and smartseq2. please read
Shan Gao. Data Analysis in Single-Cell Transcriptome Sequencing. Methods in Molecular Biology 2018, 1754: 311-326.

2. The authors need list similar work before their submission and compared their classifiers with
others, at least with a basic one, e.g. one transformer based method.

3. the datasets for training need use standard ones e.g.,
T cells: http://cf.10xgenomics.com/samples/cell-vdj/3.0.0/vdj_v1_hs_pbmc2_t/
vdj_v1_hs_pbmc2_t_clonotypes.csv
CD4 T cells: http://cf.10xgenomics.com/samples/cell-vdj/2.2.0/vdj_v1_hs_cd4_t/
vdj_v1_hs_cd4_t_clonotypes.csv
CD8 T cells: http://cf.10xgenomics.com/samples/cell-vdj/2.2.0/vdj_v1_hs_cd8_t/
vdj_v1_hs_cd8_t_clonotypes.csv
the datasets the authors used are not standard datasets, as the cell types annnoted
in these datasets have been identified by other methods.

4. please provide the sample data for training in supplementary files and explain
how to train the model.

5. please use a independent dataset to test the model, i suggest use the gene-expression
matrix from this paper.
Fulong Ji, Yong Liu, Jinsong Shi, Chunxiang Liu, Siqi Fu, Heng Wang, Bingbing Ren, Dong Mi, Shan Gao and Daqing Sun.
Single-Cell Transcriptome Analysis Reveals Mesenchymal Stem Cells in Cavernous Hemangioma. Frontiers in Cell and
Developmental Biology 2022, 10: 1-8.

Experimental design

please see 1

Validity of the findings

please see 1

Additional comments

please see 1

Reviewer 2 ·

Basic reporting

The ongoing advancement of single-cell RNA-seq techniques has underscored the importance of automated cell type identification, traditionally accomplished through unsupervised clustering and subsequent manual annotation of cell clusters. LIDER, a novel tool based on deep supervised learning, addresses this need by integrating cell embedding and deep neural network classification. It is built on a stacked denoising autoencoder, a specialized form of encoder-decoder architecture, which it uses to identify cell embedding and predict cell types. In tests across eight types of single-cell data, LIDER displayed comparable or modestly superior performance against cutting-edge methods. While the paper claims it demonstrates robustness to batch effects, no data is shown for this.

The language used is clear and professional throughout.

Sufficient background and context are provided.

No data at all is presented to support the conclusions of the section “LIDER suggests comparable robust to batch effect”. This data absolutely needs to be presented.

Experimental design

The research question is well defined and within the aims and scope of the journal: is it possible to use deep supervised learning for better cell type classification? Methods are presented with great detail and the code is available on Github.

Validity of the findings

The main comment on the paper is the comparison with Seurat. First, Seurat is not a tool designed for cell labeling and classification. It provides clustering and neighbor identification algorithms, but it is not a cell-type identification tool. So the comparison is inadequate. Second, the number of clusters identified by Seurat is a function of the “resolution” parameter. This is an arbitrary, user-provided parameter. All the conclusions from Figure 4 are based on this comparison, which would have had a very different outcome if the resolution parameter of Seurat had been adequately adjusted. If they still wish to compare to Seurat they should optimize the resolution parameter to obtain similar numbers of clusters to LIDER or Truth (i.e. in Figure 4A both Truth and Lider have 9 clusters, but the Seurat resolution parameter is inadequately tuned, and therefore results in only 4 clusters being obtained).

No data at all is presented to support the conclusions of the section “LIDER suggests comparable robust to batch effect”. This data absolutely needs to be presented.

---

## Round 0.2 · accepted · Accept

Dear Dr. Tang and colleagues:

Thanks for revising your manuscript based on the concerns raised by the reviewers. I now believe that your manuscript is suitable for publication. Congratulations! I look forward to seeing this work in print, and I anticipate it being an important resource for groups studying automatic cell type identification. Thanks again for choosing PeerJ to publish such important work.

Best,

-joe

·

Basic reporting

Authors have responded to my comments adequately. I have no further comments. I suggest the journal to accept this paper for publication

Experimental design

please see 1

Validity of the findings

please see 1

Additional comments

please see 1

Reviewer 2 ·

Basic reporting

N/A, see original review

Experimental design

N/A see original review

Validity of the findings

N/A see original review

Additional comments

Concerns have been addressed.